# Design Opportunities to Reduce Waste in Operating Rooms

**Charlotte Harding** [1] , **Joren Van Loon** [1] , **Ingrid Moons** [1,2] , **Gunter De Win** [3,4] and **Els Du Bois** [1,*]

1 Department of Product Development, Faculty of Design Sciences, University of Antwerp, 2000 Antwerp, Belgium; charlotte.harding@uantwerpen.be (C.H.); joren.vanloon@uantwerpen.be (J.V.L.); ingrid.moons@uantwerpen.be (I.M.)
2 Department of Marketing, Faculty of Business Economics, University of Antwerp, 2000 Antwerp, Belgium
3 Antwerp Surgical Training, Anatomy and Research Center (ASTARC), Faculty of Medicine and Health Sciences, University of Antwerp, 2000 Antwerp, Belgium; gunter.dewin@uza.be
4 Department of Urology, University Hospital Antwerp, 2650 Edegem, Belgium
* Correspondence: els.dubois@uantwerpen.be

**Abstract:** While taking care of the population's health, hospitals generate mountains of waste, which in turn causes a hazard to the environment of the population. The operating room is responsible for a disproportionately big amount of hospital waste. This research aims to investigate waste creation in the operating room in order to identify design opportunities to support waste reduction according to the circular economy. Eight observations and five expert interviews were conducted in a large sized hospital. The hospital's waste infrastructure, management, and sterilization department were mapped out. Findings are that washable towels and operation instruments are reused; paper, cardboard, and specific fabric are being recycled; and (non-)hazardous medical waste is being incinerated. Observation results and literature findings are largely comparable, stating that covering sheets of the operation bed, sterile clothing, sterile packaging, and department-specific products are as well the most used and discarded. The research also identified two waste hotspots: the logistical packaging (tertiary, secondary, and primary) of products and incorrect sorting between hazardous and non-hazardous medical waste. Design opportunities include optimization of recycling and increased use of reusables. Reuse is the preferred method, more specifically by exploring the possibilities of reuse of textiles, consumables, and packaging.

**Keywords:** design for sustainability; operating room; design for a circular economy; plastics; waste management; reuse; hospital waste; healthcare waste

## 1. Introduction

Hospitals have a significant environmental impact due to their size, energy-intensive processes, consumption of resources, and waste creation [1]. The increasing volume of medical waste is a growing problem that is currently often overlooked [2]. The escalation of waste is related to the aging of the population that needs more care as well as to the expansive use of single-use products [3]. Recently, the volume of medical waste also increased as a consequence of the worldwide COVID-19 pandemic and the phenomenal need for personal protective material. Moreover, in the first stages of the pandemic, the world was also hit by a shortage of protective equipment, and most recently a shortage of syringes and flacons has bogged down the roll-out of COVID-19 vaccination campaigns.

Healthcare facilities in the United States dispose of more than 4 billion pounds of waste annually [4], of which 20–33% is produced in the operating room (OR) [5,6]. Each surgery results in large amounts of waste, which mainly consists of single-use products and sterile packaging. To put this into perspective, a routine operation at a hospital produces more waste than a family of four produces in an entire week [4]. The usage of these single-use products and sterile packaging originates from sterile regulations to prevent the risk of contamination. Single use is considered to be essential to guarantee the critical hygiene

aspects and to reduce the risk of contamination, poisoning, or injuring by unmanaged (hazardous) waste [7].

The research is executed keeping in mind the probable contradiction in ethics and morality between the environmental urge to reduce the impact of hospitals and the responsibility to reduce any risk of contamination [8]. In addition, the assurance of continued availability of various products, such as personal protective equipment (PPE), to ensure safe medical treatment, should also be taken into account. Nevertheless, the usage of single-use plastic, which is the current established way to reduce risks, might not be the only possible solution. However, research is needed to investigate the consequences of new solutions that can combine environmental and human protection.

As this research was conducted in the Flemish region of Belgium, we did consider the Flemish regulations on waste management that are defined within the VLAREMA regulations [9], more specifically Art. 1.2.1.51° and Art. 5.2.3.1.§1. Here, hazardous medical waste (HMW) needs to be collected separately and incinerated at a high temperature (i.e., potential contamination of infectious diseases, sharp objects such as needles, all blood and blood derivates, anatomical waste, pathological waste, specific infected artificial organs). Non-hazardous medical waste (NHMW) includes body fluids, bandages, tissues, disposables, pads, catheters, empty blood bags, probes, syringes without needles, empty intravenous products, etc. The VLAREMA also mentions that in case of doubt, waste should always be considered HMW [9]. Nevertheless, due to the uncertainty on the hazardousness of waste, much waste is discarded incorrectly and therefore also burned unnecessarily [10].

The World Health Organization (WHO) states that hospital waste is 85% NHMW and 15% HMW, of which 10% is infectious waste and 5% non-infectious but hazardous waste [11]. In the OR, 90% of NHMW is incorrectly sorted as HMW (infectious or hazardous) [12]. This incorrect sorting results in unnecessary high costs for waste processing due to the specific need for high temperature incineration [13]. Waste from the OR consists of 39% textile, 26% different types of plastics, 7% paper and carton, and 27% of a mixture of metal, glass, sharp products, anesthetic gasses, medication, and anatomical or pathological waste [14]. The hospital's purchase and sterilization departments observe a tendency towards a growing amount of single-use products, causing an increase in the plastic waste percentage [15]. The most-used plastics in the OR are polypropylene (PP), polyethylene (PE), copolymers, polyurethane (PU), and polyvinylchloride (PVC) [16,17]. Of these materials, in theory, PP, PE, and PVC can be recycled with existing technology [16]. Another remarkable aspect found in the literature is that 80% of the waste in the OR is generated before the patient arrives [3,18,19], which is consequently always non-infectious as long as it is not mixed with other waste.

### 1.1. Opportunities for Improvements

The advantage of reducing medical waste is not only a reduction of financial cost, but also a reduction of the environmental cost. From a circular economy perspective, the reuse of products would avoid repeat purchase of new materials and allow the initial purchase price to be spread over multiple uses [20]. For the environment, resource efficiency is an important aspect, as is the reduction in emissions during production and transportation [21]. Moreover, the health sector serves as a role model to promote public health. Hospitals could be setting an example by reducing waste and single-use plastic consumption and motivate other organizations and people to do the same. The size of the health sector is massive, so moving towards a more sustainable situation would have a substantial impact [22].

In the literature, various interventions are discussed to make the OR more sustainable [3,23]. Based on circular economy principles [20], specific interventions for the OR context can be described based on the aim to reduce, reuse, or recycle. Reduction of waste can be achieved by preventing the opening of unnecessary products. Reduction of infectious and hazardous waste can also be achieved by means of better sorting in the

OR. Reuse as well as lifetime extension of products (or product components) are critical to avoid waste generation. Sorting waste also offers an excellent opportunity to retain the value of materials by means of recycling. However, as proposed by [24], a combination of education and encouragement of the staff is crucial to achieve this. It should be noted that recycling is considered to be least preferable due to the energy and actions needed [20,23]. However, it is obviously better than incinerating or dumping materials. All products made for medical applications have to be developed from primary resources, which disqualifies recycling for the same purpose [25].

Despite the increasing willingness and enthusiasm to implement sustainable interventions, different factors interfere with their implementation. Based on the literature [3,15,17,24,26–31], the following categories of obstructive factors could be identified:

1. Legislation that is formulated in a manner that excludes any possible danger and also complicates innovation;
2. Lack of instructions regarding the reduction or reuse and lack of sustainability policy for the OR;
3. Perception of sterility for single use;
4. Concern on the extra workload, time, and complexity of different types of plastics: Reuse requires additional cleaning, maintenance, and sometimes sterilization of products. Sorting (to enable recycling) requires multiple actions to separate the waste amongst multiple bins;
5. Lack of understanding of the staff regarding optimal sorting or regarding the difference between HMW and NHMW, which results in cautious actions, i.e., discarding as HMW;
6. Negative staff attitudes, intentional or unintentional, often because of the overload of activities and stress to perform optimally in a surgery;
7. Fear of reprimand: The potential effect of improper cleaned or non-effectively sterilized material or reduced sterility of material can be disastrous, and people do not want to be made responsible for it. Consequently, to avoid inappropriate discarded waste, for example, in case of doubt, the less dangerous option is preferred;
8. Bad labelling increases the difficulty of understanding how to discard;
9. Lack of sorting facilities, i.e., not all materials can be collected separately for reuse or recycling; and
10. No link between purchase and waste in cost of products, so there is no incentive to activate the search for possible alternatives.

### 1.2. Research Aim

Building further upon the identification of the different factors that obstruct the increase of sustainable actions in hospitals, this paper reports on a research project that aims to identify the various design opportunities to support reduction of waste in operating rooms. Various sub-research questions (RQs) were formulated to understand the current context, the motivations of the staff, and the other important decision-influencing parameters such as costs and organization. Based on this information, design opportunities can be identified as a reaction to deal with the previously identified obstructions to enable reduction, reuse, and recycling of products in an OR context.

Sub-questions to understand the current situation of consumables used include:

- RQ1. What is the organization (rules, norms, habits, etc.) of the OR regarding hygiene and sterility?
- RQ2. In what ways do the operating rooms differ/resemble each other (infrastructure perspective)? How is the infrastructural organization of an OR determined?
- RQ3. Which stakeholders are involved regarding the consumables of the OR?
- RQ4. What type of products are consumed during a surgery?

Sub-questions focusing on waste management and reuse include:

- RQ5. How is sorting organized during a surgery in the OR?

- RQ6. How is waste management communicated in the OR?
- RQ7. How much additional space is available for extra sorting in the OR (user perspective)?
- RQ8. How is the logistics process of waste management and product reuse organized from the viewpoint of infection prevention (including what happens to waste after it leaves the hospital)?

## 2. Materials and Methods

In order to get an answer to all research questions, two types of research activities were executed: (i) Eight observations of operating room waste handling took place to provide an answer to RQs 1–7, and (ii) five expert interviews with other stakeholders in the hospital were executed in order to formulate insights concerning RQs 1, 3, and 6–8. Based on these research activities, the waste system of the hospital could be mapped out in order to identify the opportunities for interventions.

The research was limited to the context of a single hospital. In this case the Antwerp University Hospital (UZA) was chosen as the context of inquiry. UZA has experienced a constant growth in recent years. Both the number of beds and the number of staff has grown significantly. Currently, the hospital accounts for 573 beds in 27 nursing units. Yearly, more than 27,846 patients spend the night there. In addition, in the 38 specialized medical services, more than 660,000 clinical appointments are made each year. The hospital has around 2800 employees. In 2018, 18,364 surgeries were carried out, with an average of 50 per day. UZA is considered a relevant case as it is a relatively large representative hospital for the Flemish situation that has, due to its connection with the university, an open mindset for innovation and improvement.

### 2.1. Observations

In total, 8 surgeries were observed, with a variation in the duration of the surgery itself between 1 h and 6 h. All surgeries were done within the department of urology, but considered different approaches: open surgery, laparoscopy, robotic surgery, and endoscopic surgery. Urological surgeries were considered to be relevant as they have a wide variety of techniques used, and they vary largely in duration and diversity of interventions. A surgery observation was used as the entity of investigation as it shows a large variation in sub entities, i.e., products used, staff, use of specific OR conditions.

Before the actual observations started, a first exploration of the OR context was conducted to optimize the observation research design. During this pre-test, for one hour the observer watched what happened in the OR. Based on the findings from this exploration, the actual observation structure was set up, and a detailed template was developed that supported and facilitated the final observation.

The template contained the general information on the surgery and the OR, as well as the additional feedback that was given by the staff during the observation. The observation of waste monitoring was the most important topic in this protocol. Therefore, the observation guiding list was based on the waste list for medical waste, documented by OVAM [32], the official Flemish Waste Society. In the template, the waste products were put in a table, with a second column to count the consumption per waste bin that was used and a last column to fill in additional remarks. Based on this inventory list per observation, an (Excel) overview was made reporting how many items were discarded per type of product and in which fraction they were discarded. To complete the data collection based on these observations, additional notes were made and photos were taken to allow sufficient data collection. In the pictures, patients were unidentifiable and all identifiable staff explicitly gave their permission to take their picture.

Table 1 describes the researcher's phases in the observation flow during the whole journey of the surgery in the OR. As it is known that 80% of waste is produced before the patient enters the OR [3,18,19], observations started when the staff started to prepare the OR. During the preparation of the OR, a small floorplan of the OR was drawn and

the waste bins were located on this plan. Furthermore, information was gathered about the surgery and instruments used. Next, quick questions were asked of the nursing staff regarding both used products and waste creation. Packaging and discarded consumables were counted using the list in the template. From the moment the patient arrived, the researcher kept a low profile and continued to count and register discarded items. The observation ended after the OR was cleaned up and ready for preparation of the next surgery. During cleanup, the researcher was focused on counting the consumables that were thrown away. Throughout the entire observation, the researcher was able to briefly ask for additional information about the surgery and products used, as long as the surgery was not interrupted.

**Table 1.** Observation method.

| Phases of a Surgery | Researcher's Actions |
|---|---|
| Preparation of the OR | Draw floor plan<br>Locate waste bins<br>Gather information about surgery and instruments used<br>Count discarded items with list |
| Patient arrives in the OR | Count discarded items with list |
| Cleanup of the OR | Count discarded items with list |

*2.2. Expert Interviews*

Products pass several stages in the hospital in which they are treated differently. Three main phases can be identified: (i) a pre-usage phase, when the products are stored in the hospitals and distributed to the operating rooms; (ii) the usage phase, when the products are used in the operating room; and lastly (iii), after usage, when the products are discarded in different manners and the waste is treated according to the procedures. This latter phase can also include cleaning and sterilization for reuse. To complete our insights on this product journey, for each phase in this journey, in-depth interviews were organized. Five experts were interviewed (n = 5) to be able to map the complete journey of products throughout the hospital, before and after leaving the OR. The experts were representatives of a specific department of the hospital and had decision-making power regarding the journey of the products. The selected relevant departments were the logistics department (to understand the pre-usage phase), the cleaning and waste management department (to understand the path of the discarded products), and the sterilization department (to understand the path of product reuse).

The head of the logistics department (n = 1) was interviewed to understand the pre-usage phase in the hospital. Interview topics were structured around the flow of product delivery, product storage, and distribution towards the ORs. Specific questions were asked about the types of packaging used.

To understand the path of the discarded, single-use products, the experts we interviewed included the heads of cleaning and waste management of the UZA hospital (n = 2) and one representative of the intermediary recycling partner (n = 1). Regarding cleaning and waste management, the interviewed duo was altogether coordinating 150 staff members to guarantee hospital cleanliness and hygiene. More information was requested about the specific process that each type of waste follows after leaving the OR. This interview was conducted face-to-face and included a guided tour through the hospital sorting facilities. The intermediary recycling partner was contacted by phone to get more information about a running recycling project.

To understand the path of the products that are cleaned and reused, the sterilization department of the UZA hospital was contacted (n = 1). A face-to-face interview was conducted with the head of department (n = 1) and a guided tour was taken to see the sterilization department. Specific questions were asked regarding which products and

instruments could be/were sterilized and reused and which ones were not. In addition, insights were gained regarding what the specific sterilization methods used are.

## 3. Results

### 3.1. Findings from Observations

Waste produced during a surgery originated from two different activities: (i) waste from the execution of the surgery, containing laminate packaging, gloves, single-use sheets, and gowns; and (ii) waste from anesthesia, including needles, intravenous bags, and glass vials. As the observations were only executed by a single researcher, the waste generated by anesthesia was not taken into account in this research project. This does not mean that this waste stream is not relevant, as anesthesia is responsible for 25% of the waste generated in the OR [17,24].

Four different waste streams were identified. A white bag is used for textiles (mainly towels). All HMW and sharp objects are collected in yellow bins. For sharp products, special needle containers are foreseen with a volume of 10 L. For other HMW, 60-L containers are available. A green bag is used to collect a specific type of fabric made of polypropylene (further referred to as PP-fabric) that is used to pack the sterilized instruments. Lastly, a blue bag is used for the remaining fraction (NHMW). Small versions of these blue bags are also used in metal bins on wheels. In addition, in the entrance and public part of the hospital these blue bags are used to collect all waste. All bags are positioned in holders on wheels to be positioned as desired for the surgery. Furthermore, outside of the OR, mostly in the storage area, a fifth waste stream container is available for paper and cardboard.

Operation instruments that are reused are placed in a metal basket to be sterilized. The content of these baskets is adjusted to the preferences of the surgeon and the type of surgery. The baskets are wrapped in the PP-fabrics, which should be sorted in the green bag. The packet is sealed with tape and labelled with a sticker. A particular opening procedure is used to ensure a sterile handover of the instruments.

Based on the inventory list of the observations, the most used and discarded products were (i) covering sheets for the operation bed: table cloths, belly bands, top sheets, tack cloths, etc.; (ii) sterile clothing such as gowns, gloves, face masks, clogs, and hats; (iii) sterile packaging laminate bags with paper or Tyvek; and (iv) department-specific products such as catheters, probes, ureteroscopes, urine bags, stents, and coagulation devices. These results are greatly comparable to what was discussed by other authors who referred to products such as surgeon single-use products, personal protective material, coverings, and plastic packaging [3,4,19,33]. The need for sterility demands a high level of covering and packaging that is currently provided in single-use laminate bags and other single-use products.

During the observation, additional questions were asked of the OR staff to understand their perspective on the sustainability of the OR. Most reactions were related to the following:

- The number of containers needed for sorting: Additional sorting is achievable, according to multiple OR staff (regarding time and effort), but the infrastructure was seen as the highest problem, i.e., the lack of space to put additional sorting bins.
- Experience in other places: One staff member had experience in a Spanish hospital, where at that time textile table covers were still used in the OR that were washed and sterilized. Nevertheless, she considered this to be not clean or hygienic. This was in big contrast with the other reused sterilized products that are currently used in the OR, where this perception did not appear.
- Collection of non-used material: Often in a set (= package of various products used for a specified procedure), different products are not used. Currently, they are thrown away. Recently, an experiment was done using a cart in the corridor to put unused clean products (for which the sterility cannot be guaranteed and so cannot be used anymore for human surgery). Colleagues could take these products for personal usage or to donate to vets or other less critical situations. People argued that some specific

textiles are used to clean cars or product containers that are used for personal storage. This type of product recycling was also described by Lausten [31], where unused items were donated to developing countries.

During the observations, a ground plan was drawn of the ORs. In total, the observations took place in four different ORs. In Figure 1, the four ground plans are shown, with the intent to get insights into the location of the different waste bins. In general, the ORs were relatively comparable. In each situation, blue (NHMW) and yellow (HMW) containers are placed at the head of the surgical bed. Here, the anesthetist is seated, which enables him or her to discard waste without leaving the patient. Next, a combination of white (textiles), green (PP-fabric), and blue (HMW) recipients is situated near the storage area. This enables the nurses who prepare the OR to discard the packaging properly. As mentioned before, the paper and cardboard bin is most of the time present in the storage area. It should be mentioned that this composition can vary thoroughly in other ORs as the bins can be moved freely in the room or specific waste fractions can be missing due to various reasons. Nevertheless, mapping the infrastructure in the space is valuable to avoid failing with specific interventions due to a lack of insight into the infrastructure.

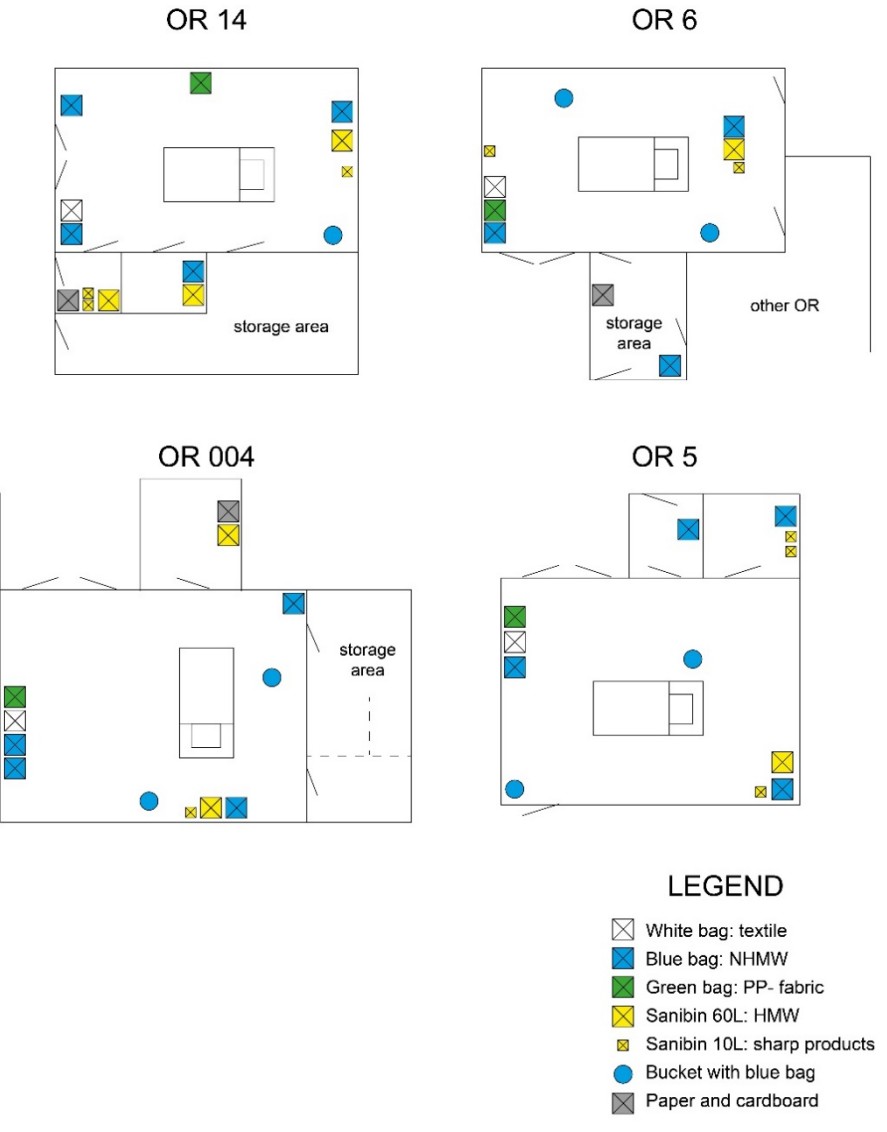

**Figure 1.** Waste bin arrangement in the different operating rooms.

*3.2. Findings from Expert Interviews on Logistics and Waste Management*

After an order from the purchasing department, the purchased products that are needed in the ORs are delivered to the hospitals. Delivered goods are stored in the internal warehouse and are distributed amongst all departments in the hospital. Tertiary packaging is removed at the warehouse so that only secondary packaging is stored in the ORs' storage rooms. Only the primary packaging is taken inside the OR during the preparation of the OR for a surgery. Every hospital is obligated to have a waste stream management list. Based on this list, the sorting instructions for the hospital staff are detailed and communicated. All waste bags are tagged with OR room number, date, and time to allow traceability. All waste containers and bags are collected in a waste collection point near the ORs, which is regularly emptied by the cleaning staff. The white bags with textiles are collected in a laundry cart and collected by an external laundry company (Cleanlease) to be washed and sterilized. The green bags with the PP-fabrics are picked up by an intermediary partner (Actio), where stickers and tape are manually removed. Once stickers and labels are removed, the PP-fabrics go to recycling partner (Innorem), where they are processed into plastic pellets for recycling. Unfortunately, there are still some difficulties experienced in the execution of the project. The manual processing requires strict monitoring for correct sorting: A glove or a syringe regularly gets lost between the PP-fabrics in the green bags. For the safety of its employees, "contaminated" bags are returned to UZA. These bags end up being processed with the other NHMW at the hospital, i.e., incinerated.

The yellow saniboxes (HMW), the blue bags (NHMW), and paper and cardboard are transported from the OR waste collection point to the recycling park of the hospital. There, all waste streams are separately stored. The yellow boxes are stocked in a sealed container for security reasons, as they contain hazardous waste. All waste is processed by Indaver: HMW is burned at a high temperature in a rotary kiln to reduce any risk of infection and NHMW is also burned at Indaver, but following the normal procedure with energy recuperation. Paper and cardboard are recycled. Figure 2 shows an overview of the waste flow from OR to processing.

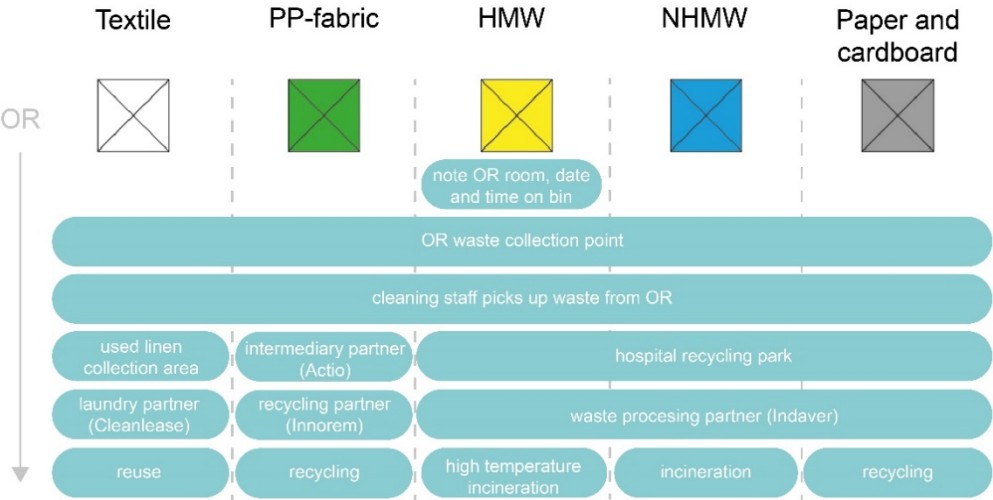

**Figure 2.** Overview of the journey of the different waste flows after being discarded in the OR.

*3.3. Findings from Expert Interview on the Central Sterilization Department*

Sterilization is required for instruments and textiles that break the skin or mucous membrane and thus come into contact with sterile tissues of the body. The sterilization techniques used at UZA are steam sterilization and $H_2O_2$ (hydrogen peroxide) or plasma sterilization. Steam sterilization is the cheaper and safer option, and thus has preference. It uses high temperatures up to 134 °C. Some products or instruments are not heat resistant and require a different technique: plasma sterilization, which uses temperatures up to

50 °C. Many frequently used products (syringes, tubes, catheters, etc.) are not sterilized for reuse because they are designed and manufactured for single use. Single-use products carry a symbol that forbids using them multiple times because safety and hygiene cannot be guaranteed. The ability to sterilize a product is a combination of design and material selection. The product must be designed so that the sterilizing agent can reach all surfaces; long hollow spaces do not pose any problem. In addition, the selected material must be resistant to cleaning agents and heat. For the central sterilization department to sterilize a product, the manufacturer of medical devices must indicate that sterilization is allowed and provide cleaning and sterilization instructions.

In the central sterilization department, medical tools and instruments are maintained (repaired), cleaned, and sterilized. This can be individual instruments or sets, which are compositions of instruments needed to perform a certain type of surgery. Instruments are packed to ensure sterility until usage by offering a barrier between the sterilized instrument and the environment. Different packaging systems are used: (i) packaging using sterilization PP-fabrics in different sizes and provided with a lot number. These can be used for both plasma and steam sterilization and are mainly used for heavy sets or instruments. An instrument is wrapped in a piece of fabric using a specific folding technique and closed with a self-adhesive tape with a passage indicator. The sterility of an object cannot be demonstrated without performing tests in which its sterility is lost. Consequently, during the sterilization process, process indicators are used. These are visible indicators that change color when a product has gone through a sterilization process. This indicator is located on the packaging of the instrument. Next, (ii) packaging in laminate bags is used for lightweight and small instruments. These are made from two different materials: paper and PP for steam sterilization, and Tyvek and PP for plasma sterilization. Similarly, lot numbers and passage indicators are used. The sealing is ensured by means of a welding machine to melt the PP in the fibers of the paper or Tyvek. (iii) Packaging in containers is used for extended sets or when the sets needs to be transported outside of hospitals. These containers are reusable packaging that need to comply with the following norms: EN 868-8, EN 16775, and EN ISO 11607-1 and -2, and are made from stainless steel to ensure a longer lifetime resisting disinfectant products and high temperatures. The containers are sealed with a process indicator. Unfortunately, it is not possible to use these reusable containers for all products as they take a lot of space to store and ORs are short on storage space.

### 3.4. Synthesis of the Findings

A visual overview (Figure 3) was created to gain insights into the journey that products follow to and from the OR. The arrows in the figure represent the different waste streams from the OR to the discussed (external) stakeholders.

Closing the loop by means of recycling is not possible, as medical products should be produced from virgin materials. Consequently, two waste hotspots were identified based on the synthesis overview: (i) The combination of the usage of tertiary, secondary, and primary packaging should be taken into account in the amount of waste per product, and (ii) due to uncertainty and risk reduction, the waste sorting between HMW and NHMW is not done in the correct way.

Regarding reuse, three product categories should be considered based on products that were observed frequently. These products are excessively used during surgeries and offer a significant volume to be replaced by reusable alternatives:

1. Textiles: table covers, patient overlays, surgical gowns, caps, facemasks, etc. This product category is responsible for high volumes of material in the blue waste bins. In the past, the predecessors of these products were natural reusable textiles such as linen or cotton.
2. Consumables: syringes, plastic forceps, tubes, plastic trays, catheters, etc. These are small, simple products that are used in different types of surgical procedures. Currently, these are all made for single use, so they should not be sterilized. However,

thanks to their simple design (shape), they could be perfectly cleaned and sterilized if the manufacturer provided this in the design (and material).

3.  Packaging: laminate bags, sterilization wraps, plastic trays, etc. It is relatively easy to separate the PP layer and paper layer of laminate bags during disposal. It is unfortunate that reuse of products (through sterilization) results in the creation of another type of waste. It would be much better if products could be sterilized in reusable packaging.

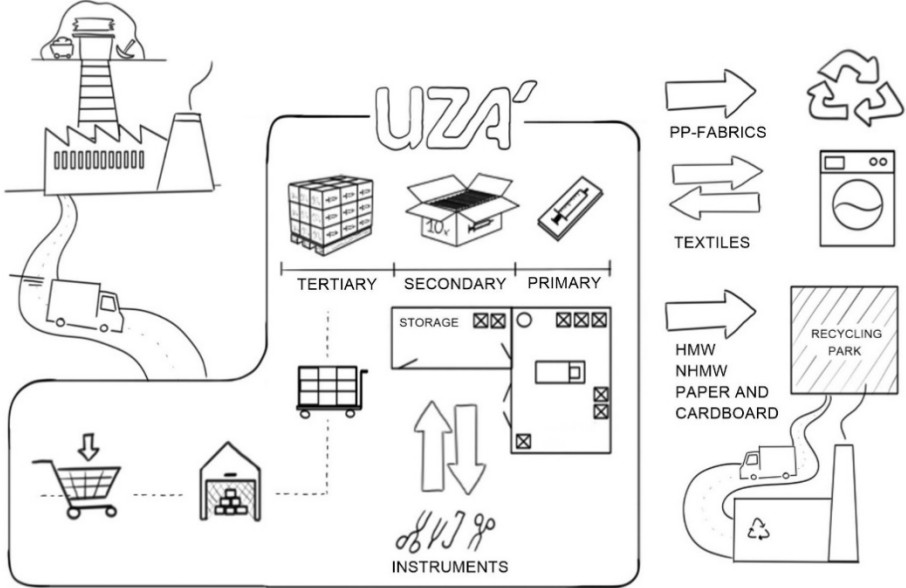

**Figure 3.** Overview of the journey of different types of products throughout the OR.

## 4. Discussion

By means of an in-depth exploration of the extended research team (n = 9) based on the results of the analysis, various opportunities were identified to reduce the impact of the waste generated by ORs. The opportunities can be organized into two perspectives: recycling optimization opportunities and reuse opportunities. Preference between reuse and recycling can be made based on the model of circular economy, where extending lifetime is considered to have a higher value than recycling (reuse of materials) or based on older reasonings such as Lansink's ladder [34], which actually results in the same conclusion that reuse should be preferred over recycling. However, it should be mentioned that the two perspectives should be combined to achieve an optimal solution, as reuse is not always viable.

Reuse: Opportunities for reuse after sufficient cleaning, decontamination, and sterilization were found in (i) alternatives for tertiary and secondary packaging, where empty packaging can be returned by the next delivery truck of the specific company. (ii) Reuse of protective clothes, as disposable surgical gloves and gowns contribute disproportionately to the amount of surgical waste [4]. The ecological and economic impact of reusable and disposable surgical gowns has been researched in different comparisons [35–37]. During the observation, two staff members used a reusable cap. All other cloths used to protect the bed, patient, and staff were made from sterile single-use materials that were discarded through the NHMW or HMW bins afterwards. (iii) The extension of reusable sterilization packaging is considered an interesting opportunity, as this is already done for large and on-the-go packaging. (iv) In particular, the PP-fabric packaging that is currently discarded separately should be considered if reuse can be achieved instead of regranulation for other purposes. (v) The composition of sets should be reconsidered to avoid unused products and their unnecessary costs, as was described by previous research [4,31,38–40], or should be organized in such a manner that the sterility can be guaranteed for reuse. (vi) Further-

more, different instruments are used that exist in both single use and sterilizable, reusable versions. The aim should be to use reusable instruments whenever possible. Sustainable operating room procurement has been suggested [22,23,31,33,41]. (vii) Lastly, a large opportunity exists to communicate the proof of cleanness to be able to identify the sterility of non-used products and allow future usage instead of the discarding that is currently done.

Recycling: Optimal sorting could be extended and optimized to sort out a higher diversity of materials. Recycling is something that many people practice at home, but many still struggle with developing this habit in a health care facility [31]. One of the products that was discarded the most according to the observations are laminate bags, existing in two materials: PP and paper or PP and Tyvek (which is actually polyethylene). These two layers can easily be separated and collected separately for higher value recycling. However, to achieve this the sorting units should be reconsidered to optimize discarding of products, for example by making these sorting units more readily available [6]. Simultaneously, this intervention of the sorting units should clarify the confusion amongst the risk of waste, as mentioned by [13]. One specific opportunity mentioned is the use of matching markings on the products to identify the specific waste bin. This is also done in other contexts such as festivals to optimize sorting and reduce the number of incorrectly sorted items.

From an economic perspective, the cost of introducing new systems deserves further investigation. Not all waste can be eliminated; however, there is a considerable potential to reduce waste and the related costs (definitely by optimization of sorting and thereby lowering the amount of incorrectly sorted HMW). Reduction of non-used discarded products has a direct impact on waste reduction and purchase costs. Regarding the potential benefit of replacing disposable products by reusable alternatives, the cost of waste in combination with the purchase costs of the products (plus the societal cost for the environment) could be used as an indicator to understand the value for introducing new reusable systems. An important difference that should be taken into account is the change of ownership of reusable products and the pay per usage that is often introduced, which might completely change the willingness of hospitals to pay. Lastly, the cost of building the image of a green hospital can also be an investment in reusable products and other waste reductive actions.

Lastly, it should be mentioned that the interviews and observations were executed just before the COVID-19 pandemic. We can reason upon the fact that due to the pandemic, the importance of hygiene and infection prevention has only become more important. The (perception of) risk of infection turned the transition towards a circular economy often in a negative direction, by (re-)introducing (additional) single use plastics [42]. However, on the other hand, most hospitals also face massive amounts of additional waste [43] and shortages of PPE further initiates the debate towards reuse [44,45]. As hospital regulation is built to prevent and limit any contamination, no changes of regulations were done. However, what we learned from the pandemic regarding reuse and recycling is that the involvement of third parties in the reuse and recycling value chain should be done with additional caution, as infected material is transported to an environment outside of hospitals were regulations might be less strictly followed.

## 5. Conclusions

Based on previous research, we can conclude that the waste generated in operating rooms (ORs) is significantly more than waste generated in other departments in a hospital, to the extent that the OR is responsible for up to 33% of the waste generated in a hospital. In fact, an average surgery produces more waste than an average four-person family in a whole week [3,4]. The aim of this research was to investigate from a design perspective how waste production can be reduced and which design opportunities can be identified to facilitate this. In our search for design opportunities to support OR greening, reuse was identified as a big enabler to reduce surgical waste. In addition, various recycling optimizations should be further developed to increase the value of the generated waste.

Within this research, the following opportunities to reduce the waste of ORs could be identified:

- Opportunity 1: reuse of secondary and tertiary packaging;
- Opportunity 2: reuse of textiles and protective clothing;
- Opportunity 3: reusable sterilization packaging;
- Opportunity 4: reuse of instruments;
- Opportunity 5: communication of cleanliness to enable future use;
- Opportunity 6: optimization of waste sorting by clarifying the difference between NHMW and HMW;
- Opportunity 7: use of product coding to enable efficient sorting;
- Opportunity 8: separate collection of PP and paper (or Tyvek) of the various packaging; and
- Opportunity 9: optimization of the contents of surgical sets.

Future research should explore the feasibility and practicability of reuse in the healthcare sector. Reuse has the image of being old fashioned and of less quality. Healthcare workers play an enormous role in the decision process of new products used in the OR. How can they be convinced of the benefits of reuse? Further research is needed to understand the causality of this perception of sterility and hygiene in various products (single use versus reused). Next, the economic feasibility should be defined, since price keeps playing a huge factor in the purchasing department.

Future research should be executed to translate these opportunities into executable design realizations. Furthermore, further research should also include the perspective of anesthesia.

**Author Contributions:** Conceptualization, G.D.W.; data curation, C.H.; methodology, C.H. and E.D.B.; validation, G.D.W., I.M. and J.V.L.; investigation, C.H.; resources, G.D.W.; writing—original draft preparation, C.H. and E.D.B.; writing—review and editing, G.D.W., I.M. and J.V.L.; visualization, C.H.; supervision, E.D.B.; project administration, C.H. All authors have read and agreed to the published version of the manuscript.

**Funding:** This research received no external funding.

**Data Availability Statement:** The data presented in this study are available on request from the corresponding authors.

**Acknowledgments:** We thank the UZA hospital and its personnel. In specific, we would like to thank the urology department, OR staff, logistics, sterilization, and cleaning and waste department for the pleasant cooperation and willingness to contribute to this research. In particular, we would like to thank Ann De Troetsel for making the observations in the OR possible.

**Conflicts of Interest:** The authors declare no conflict of interest.

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
