# Peer review of "Design Opportunities to Reduce Waste in Operating Rooms"

_sustainability, doi:10.3390/su13042207_

Round 1

Reviewer 1 Report

1. Thank you very much for submitting this study for review. The description of the data collection method deserves special attention, as well as  the observation itself. As a recommendation, it is possible to advise the authors to arrange the procedure for conducting observations in a short scheme, algorithm, or even instructions, so that this observation can be repeated in other conditions.
2. As for the Discussion, we would like to see a comparison of the results obtained with the results of other similar studies. Unfortunately, the results have not been compared to date.
3. In addition to recommendations, from our point of view, it is advisable to provide predicted indicators of waste reduction, try to estimate economic efficiency, the costs of introducing recommendations, as well as savings. 

Reviewer 2 Report

Although the idea presented in this paper is interesting, it is merely descriptive. In general, I miss specific data.

Other facts to take into account are:

Line 125: A dot is missing.

Lines 147-148: Explain better what RQ1-7 and RQ1:3;6-8 mean.

Line 161: Why did you take into account only urology surgeries?. Please explain how many persons use de OR each day.

Line 199: What kind of experts were they?

Line 213: Ii is not clear what n=1, n=2 etc. mean. Please explain them before using them.

Line 218: Could you provide more information about these interviews? For exemple number of questions, type of questions, number of people answerring the questions, etc.

Line 265: In my opinion, this information about other hospital is irrelevant.

Line 279: Explain what type of waste must be placed in each bin. It’s only indicated in the figure’s legend.

Line 356-357: Here the letters format changes.

Lie 370: Which amount exactly?

Reviewer 3 Report

Dear Editor: this paper is worth to be published here especially currently many countries are fighting with COVID-19, and are facing the medical protection materials supply and disposal problem. However, to reuse or recycle hospital wastes is opposite with the willingness of people in the meantime. Therefore, authors have to make more clear what is the target of this paper.  My suggest for the authors is to re-organize and keep the content of sorting system in the hospital basing on the viewpoint of infection prevention during epidemic period in this paper.

Round 2

Reviewer 3 Report

As i my first review, this article shall be published here and can focused on the sorting and separation in current situation of COVID-19. Therefore, i am still not agree the wording of "Reuse" and "Recycle".  

Round 3

Reviewer 3 Report

Dear Editor : Now it can be published as soon as possible.